# FEDERATED LEARNING WITH MATCHED AVERAGING

**Hongyi Wang** [*]
Department of Computer Sciences
University of Wisconsin-Madison
`hongyiwang@cs.wisc.edu`

**Mikhail Yurochkin**
IBM Research
MIT-IBM Watson AI Lab
`mikhail.yurochkin@ibm.com`

**Yuekai Sun**
Department of Statistics
University of Michigan
`yuekai@umich.edu`

**Dimitris Papailiopoulos**
Department of Electrical and Computer Engineering
University of Wisconsin-Madison
`dimitris@papail.io`

**Yasaman Khazaeni**
IBM Research
`yasaman.khazaeni@us.ibm.com`

## ABSTRACT

Federated learning allows edge devices to collaboratively learn a shared model while keeping the training data on device, decoupling the ability to do model training from the need to store the data in the cloud. We propose the Federated matched averaging (FedMA) algorithm designed for federated learning of modern neural network architectures *e.g.* convolutional neural networks (CNNs) and LSTMs. FedMA constructs the shared global model in a layer-wise manner by *matching and averaging* hidden elements (*i.e.* channels for convolution layers; hidden states for LSTM; neurons for fully connected layers) with similar feature extraction signatures. Our experiments indicate that FedMA not only outperforms popular state-of-the-art federated learning algorithms on deep CNN and LSTM architectures trained on real world datasets, but also reduces the overall communication burden.[1]

## 1 INTRODUCTION

Edge devices such as mobile phones, sensors in a sensor network, or vehicles have access to a wealth of data. However, due to data privacy concerns, network bandwidth limitation, and device availability, it's impractical to gather all the data from the edge devices at the data center and conduct centralized training. To address these concerns, federated learning is emerging (McMahan et al., 2017; Li et al., 2019; Smith et al., 2017; Caldas et al., 2018; Bonawitz et al., 2019; Kairouz et al., 2019) as a paradigm that allows local clients to collaboratively train a shared global model.

The typical federated learning paradigm involves two stages: (i) clients train models with their local datasets independently, and (ii) the data center gathers the locally trained models and aggregates them to obtain a shared global model. One of the standard aggregation methods is FedAvg (McMahan et al., 2017) where parameters of local models are averaged element-wise with weights proportional to sizes of the client datasets. FedProx (Sahu et al., 2018) adds a proximal term to the client cost functions, thereby limiting the impact of local updates by keeping them close to the global model. Agnostic Federated Learning (AFL) (Mohri et al., 2019), as another variant of FedAvg, optimizes a centralized distribution that is a mixture of the client distributions.

One shortcoming of FedAvg is coordinate-wise averaging of weights may have drastic detrimental effects on the performance of the averaged model and adds significantly to the communication burden. This issue arises due to the permutation invariance of neural network (NN) parameters, i.e. for

---

[*]Work performed while doing an internship at IBM Research.

[1]Code is available at `https://github.com/IBM/FedMA`

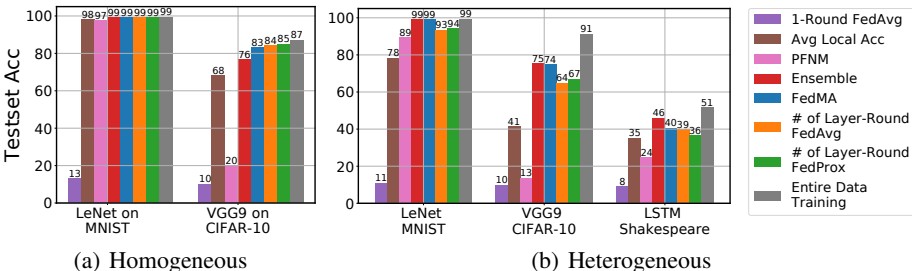

Figure 1: Comparison among various federated learning methods with limited number of communications on LeNet trained on MNIST; VGG-9 trained on CIFAR-10 dataset; LSTM trained on Shakespeare dataset over: (a) homogeneous data partition (b) heterogeneous data partition.

any given NN, there are many variants of it that only differ in the ordering of parameters. Probabilistic Federated Neural Matching (PFNM) (Yurochkin et al., 2019b) addresses this problem by matching the neurons of client NNs before averaging them. PFNM further utilizes Bayesian nonparametric methods to adapt to global model size and to heterogeneity in the data. As a result, PFNM has better performance and communication efficiency than FedAvg. Unfortunately, the method only works with simple architectures (e.g. fully connected feedforward networks).

**Our contribution** In this work, we demonstrate how PFNM can be applied to CNNs and LSTMs, but we find that it only gives very minor improvements over weight averaging. To address this issue, we propose Federated Matched Averaging (FedMA), a new layers-wise federated learning algorithm for modern CNNs and LSTMs that appeal to Bayesian nonparametric methods to adapt to heterogeniety in the data. We show empirically that FedMA not only reduces the communcations burden, but also outperforms state-of-the-art federated learning algorithms.

## 2    FEDERATED MATCHED AVERAGING OF NEURAL NETWORKS

In this section we will discuss permutation invariance classes of prominent neural network architectures and establish the appropriate notion of averaging in the parameter space of NNs. We will begin with the simplest case of a single hidden layer fully connected network, moving on to deep architectures and, finally, convolutional and recurrent architectures.

**Permutation invariance of fully connected architectures** A basic fully connected (FC) NN can be formulated as $\hat{y} = \sigma(xW_1)W_2$ (without loss of generality, biases are omitted to simplify notation), where $\sigma$ is the non-linearity (applied entry-wise). Expanding the preceding expression $\hat{y} = \sum_{i=1}^{L} W_{2,i\cdot}\sigma(\langle x, W_{1,\cdot i}\rangle)$, where $i\cdot$ and $\cdot i$ denote $i$th row and column correspondingly and $L$ is the number of hidden units. Summation is a permutation invariant operation, hence for any $\{W_1, W_2\}$ there are $L!$ practically equivalent parametrizations if this basic NN. It is then more appropriate to write

$$\hat{y} = \sigma(xW_1\Pi)\Pi^T W_2, \text{ where } \Pi \text{ is any } L \times L \text{ permutation matrix.} \tag{1}$$

Recall that permutation matrix is an orthogonal matrix that acts on rows when applied on the left and on columns when applied on the right. Suppose $\{W_1, W_2\}$ are optimal weights, then weights obtained from training on two homogeneous datasets $X_j, X_{j'}$ are $\{W_1\Pi_j, \Pi_j^T W_2\}$ and $\{W_1\Pi_{j'}, \Pi_{j'}^T W_2\}$. It is now easy to see why naive averaging in the parameter space is not appropriate: with high probability $\Pi_j \neq \Pi_{j'}$ and $(W_1\Pi_j + W_1\Pi_{j'})/2 \neq W_1\Pi$ for any $\Pi$. To meaningfully average neural networks in the weight space we should first undo the permutation $(W_1\Pi_j\Pi_j^T + W_1\Pi_{j'}\Pi_{j'}^T)/2 = W_1$.

### 2.1    MATCHED AVERAGING FORMULATION

In this section we formulate practical notion of parameter averaging under the permutation invariance. Let $w_{jl}$ be $l$th neuron learned on dataset $j$ (i.e. $l$th column of $W^{(1)}\Pi_j$ in the previous example),

$\theta_i$ denote the $i$th neuron in the global model, and $c(\cdot, \cdot)$ be an appropriate similarity function between a pair of neurons. Solution to the following optimization problem are the required permutations:

$$\min_{\{\pi_{li}^j\}} \sum_{i=1}^{L} \sum_{j,l} \min_{\theta_i} \pi_{li}^j c(w_{jl}, \theta_i) \text{ s.t. } \sum_i \pi_{li}^j = 1 \; \forall \, j, l; \; \sum_l \pi_{li}^j = 1 \; \forall \, i, j. \tag{2}$$

Then $\Pi_{jli}^T = \pi_{li}^j$ and given weights $\{W_{j,1}, W_{j,2}\}_{j=1}^J$ provided by $J$ clients, we compute the federated neural network weights $W_1 = \frac{1}{J} \sum_j W_{j,1} \Pi_j^T$ and $W_2 = \frac{1}{J} \sum_j \Pi_j W_{j,2}$. We refer to this approach as *matched averaging* due to relation of equation 2 to the maximum bipartite matching problem. We note that if $c(\cdot, \cdot)$ is squared Euclidean distance, we recover objective function similar to k-means clustering, however it has additional constraints on the "cluster assignments" $\pi_{li}^j$ necessary to ensure that they form permutation matrices. In a special case where all local neural networks and the global model are assumed to have same number of hidden neurons, solving equation 2 is equivalent to finding a Wasserstein barycenter (Agueh & Carlier, 2011) of the empirical distributions over the weights of local neural networks. Concurrent work of Singh & Jaggi (2019) explores the Wasserstein barycenter variant of equation 2.

**Solving matched averaging** Objective function in equation 2 can be optimized using an iterative procedure: applying the Hungarian matching algorithm (Kuhn, 1955) to find permutation $\{\pi_{li}^{j'}\}_{l,i}$ corresponding to dataset $j'$, holding other permutations $\{\pi_{li}^j\}_{l,i,j \neq j'}$ fixed and iterating over the datasets. Important aspect of Federated Learning that we should consider here is the data heterogeneity. Every client will learn a collection of feature extractors, i.e. neural network weights, representing their individual data modality. As a consequence, feature extractors learned across clients may overlap only *partially*. To account for this we allow the size of the global model $L$ to be an unknown variable satisfying $\max_j L_j \leq L \leq \sum_j L_j$ where $L_j$ is the number of neurons learned from dataset $j$. That is, global model is at least as big as the largest of the local models and at most as big as the concatenation of all the local models. Next we show that matched averaging with adaptive global model size remains amendable to iterative Hungarian algorithm with a special cost.

At each iteration, given current estimates of $\{\pi_{li}^j\}_{l,i,j \neq j'}$, we find a corresponding global model $\{\theta_i = \arg\min_{\theta_i} \sum_{j \neq j',l} \pi_{li}^j c(w_{jl}, \theta_i)\}_{i=1}^{L}$ (this is typically a closed-form expression or a simple optimization sub-problem, e.g. a mean if $c(\cdot, \cdot)$ is Euclidean) and then we will use Hungarian algorithm to match this global model to neurons $\{w_{j'l}\}_{l=1}^{L_{j'}}$ of the dataset $j'$ to obtain a new global model with $L \leq L' \leq L + L_{j'}$ neurons. Due to data heterogeneity, local model $j'$ may have neurons not present in the global model built from other local models, therefore we want to avoid "poor" matches by saying that if the optimal match has cost larger than some threshold value $\epsilon$, instead of matching we create a new global neuron from the corresponding local one. We also want a modest size global model and therefore penalize its size with some increasing function $f(L')$. This intuition is formalized in the following *extended* maximum bipartite matching formulation:

$$\min_{\{\pi_{li}^{j'}\}_{l,i}} \sum_{i=1}^{L+L_{j'}} \sum_{j=1}^{L_{j'}} \pi_{li}^{j'} C_{li}^{j'} \text{ s.t. } \sum_i \pi_{li}^{j'} = 1 \; \forall \, l; \; \sum_l \pi_{li}^j \in \{0,1\} \; \forall \, i, \text{ where}$$

$$C_{li}^{j'} = \begin{cases} c(w_{j'l}, \theta_i), & i \leq L \\ \epsilon + f(i), & L < i \leq L + L_{j'}. \end{cases} \tag{3}$$

The size of the new global model is then $L' = \max\{i : \pi_{li}^{j'} = 1, \; l = 1, \ldots, L_{j'}\}$. We note some technical details: after the optimization is done, each corresponding $\Pi_j^T$ is of size $L_j \times L$ and is not a permutation matrix in a classical sense when $L_j \neq L$. Its functionality is however similar: taking matrix product with a weight matrix $W_j^{(1)} \Pi_j^T$ implies permuting the weights to align with weights learned on the other datasets and padding with "dummy" neurons having zero weights (alternatively we can pad weights $W_j^{(1)}$ first and complete $\Pi_j^T$ with missing rows to recover a proper permutation matrix). This "dummy" neurons should also be discounted when taking average. Without loss of generality, in the subsequent presentation we will ignore these technicalities to simplify the notation.

To complete the matched averaging optimization procedure it remains to specify similarity $c(\cdot, \cdot)$, threshold $\epsilon$ and model size penalty $f(\cdot)$. Yurochkin et al. (2019a;b;c) studied fusion, i.e. aggregation,

of model parameters in a range of applications. The most relevant to our setting is Probabilistic Federated Neural Matching (PFNM) (Yurochkin et al., 2019b). They arrived at a special case of equation 3 to compute maximum a posteriori estimate (MAP) of their Bayesian nonparametric model based on the Beta-Bernoulli process (BBP) (Thibaux & Jordan, 2007), where similarity $c(w_{jl}, \theta_i)$ is the corresponding posterior probability of $j$th client neuron $l$ generated from a Gaussian with mean $\theta_i$, and $\epsilon$ and $f(\cdot)$ are guided by the Indian Buffet Process prior (Ghahramani & Griffiths, 2005). Instead of making heuristic choices, this formulation provides a model-based specification of equation 3. We refer to a procedure for solving equation 2 with the setup from Yurochkin et al. (2019b) as BBP-MAP. We note that their PFNM is only applicable to fully connected architectures limiting its practicality. Our *matched averaging* perspective allows to formulate averaging of widely used architectures such as CNNs and LSTMs as instances of equation 2 and utilize the BBP-MAP as a solver.

## 2.2 PERMUTATION INVARIANCE OF KEY ARCHITECTURES

Before moving onto the convolutional and recurrent architectures, we discuss permutation invariance in *deep* fully connected networks and corresponding matched averaging approach. We will utilize this as a building block for handling LSTMs and CNN architectures such as VGG (Simonyan & Zisserman, 2014) widely used in practice.

**Permutation invariance of deep FCs**  We extend equation 1 to recursively define deep FC network:

$$x_n = \sigma(x_{n-1} \Pi_{n-1}^T W_n \Pi_n), \tag{4}$$

where $n = 1, \ldots, N$ is the layer index, $\Pi_0$ is identity indicating non-ambiguity in the ordering of input features $x = x_0$ and $\Pi_N$ is identity for the same in output classes. Conventionally $\sigma(\cdot)$ is any non-linearity except for $\hat{y} = x_N$ where it is the identity function (or softmax if we want probabilities instead of logits). When $N = 2$, we recover a single hidden layer variant from equation 1. To perform matched averaging of deep FCs obtained from $J$ clients we need to find permutations for every layer of every client. Unfortunately, permutations within any consecutive pair of intermediate layers are coupled leading to a NP-hard combinatorial optimization problem. Instead we consider recursive (in layers) matched averaging formulation. Suppose we have $\{\Pi_{j,n-1}\}$, then plugging $\{\Pi_{j,n-1}^T W_{j,n}\}$ into equation 2 we find $\{\Pi_{j,n}\}$ and move onto next layer. The recursion base for this procedure is $\{\Pi_{j,0}\}$, which we know is an identity permutation for any $j$.

**Permutation invariance of CNNs**  The key observation in understanding permutation invariance of CNNs is that instead of neurons, channels define the invariance. To be more concrete, let $\mathrm{Conv}(x, W)$ define convolutional operation on input $x$ with weights $W \in \mathbb{R}^{C^{in} \times w \times h \times C^{out}}$, where $C^{in}$, $C^{out}$ are the numbers of input/output channels and $w, h$ are the width and height of the filters. Applying any permutation to the output dimension of the weights and then same permutation to the input channel dimension of the subsequent layer will not change the corresponding CNN's forward pass. Analogous to equation 4 we can write:

$$x_n = \sigma(\mathrm{Conv}(x_{n-1}, \Pi_{n-1}^T W_n \Pi_n)). \tag{5}$$

Note that this formulation permits pooling operations as those act within channels. To apply matched averaging for the $n$th CNN layer we form inputs to equation 2 as $\{w_{jl} \in \mathbb{R}^D\}_{l=1}^{C_n^{out}}$, $j = 1, \ldots, J$, where $D$ is the flattened $C_n^{in} \times w \times h$ dimension of $\Pi_{j,n-1}^T W_{j,n}$. This result can be alternatively derived taking the IM2COL perspective. Similar to FCs, we can recursively perform matched averaging on deep CNNs. The immediate consequence of our result is the extension of PFNM (Yurochkin et al., 2019b) to CNNs. Empirically, see Figure 1, we found that this extension performs well on MNIST with a simpler CNN architecture such as LeNet (LeCun et al., 1998) (4 layers) and significantly outperforms coordinate-wise weight averaging (1 round FedAvg). However, it breaks down for more complex architecture, e.g. VGG-9 (Simonyan & Zisserman, 2014) (9 layers), needed to obtain good quality prediction on a more challenging CIFAR-10.

**Permutation invariance of LSTMs**  Permutation invariance in the recurrent architectures is associated with the ordering of the hidden states. At a first glance it appears similar to fully connected architecture, however the important difference is associated with the permutation invariance

of the hidden-to-hidden weights $H \in \mathbb{R}^{L \times L}$, where $L$ is the number of hidden states. In particular, permutation of the hidden states affects *both* rows and columns of $H$. Consider a basic RNN $h_t = \sigma(h_{t-1}H + x_t W)$, where $W$ are the input-to-hidden weights. To account for the permutation invariance of the hidden states, we notice that dimensions of $h_t$ should be permuted in the same way for any $t$, hence

$$h_t = \sigma(h_{t-1}\Pi^T H\Pi + x_t W\Pi). \tag{6}$$

To match RNNs, the basic sub-problem is to align hidden-to-hidden weights of two clients with Euclidean similarity, which requires minimizing $\|\Pi^T H_j\Pi - H_{j'}\|_2^2$ over permutations $\Pi$. This is a quadratic assignment problem known to be NP-hard (Loiola et al., 2007). Fortunately, the same permutation appears in an already familiar context of input-to-hidden matching of $W\Pi$. Our matched averaging RNN solution is to utilize equation 2 plugging-in input-to-hidden weights $\{W_j\}$ to find $\{\Pi_j\}$. Then federated hidden-to-hidden weights are computed as $H = \frac{1}{J}\sum_j \Pi_j H_h \Pi_j^T$ and input-to-hidden weights are computed as before. We note that Gromov-Wasserstein distance (Gromov, 2007) from the optimal transport literature corresponds to a similar quadratic assignment problem. It may be possible to incorporate hidden-to-hidden weights $H$ into the matching algorithm by exploring connections to approximate algorithms for computing Gromov-Wasserstein barycenter (Peyré et al., 2016). We leave this possibility for future work.

To finalize matched averaging of LSTMs, we discuss several specifics of the architecture. LSTMs have multiple cell states, each having its individual hidden-to-hidden and input-to-hidden weights. In out matched averaging we stack input-to-hidden weights into $SD \times L$ weight matrix ($S$ is the number of cell states; $D$ is input dimension and $L$ is the number of hidden states) when computing the permutation matrices and then average all weights as described previously. LSTMs also often have an embedding layer, which we handle like a fully connected layer. Finally, we process deep LSTMs in the recursive manner similar to deep FCs.

## 2.3 Federated Matched Averaging (FedMA) algorithm

Defining the permutation invariance classes of CNNs and LSTMs allows us to extend PFNM (Yurochkin et al., 2019b) to these architectures, however our empirical study in Figure 1 demonstrates that such extension fails on deep architectures necessary to solve more complex tasks. Our results suggest that recursive handling of layers with matched averaging may entail poor overall solution. To alleviate this problem and utilize the strength of matched averaging on "shallow" architectures, we propose the following layer-wise matching scheme. First, data center gathers *only* the weights of the first layers from the clients and performs one-layer matching described previously to obtain the first layer weights of the federated model. Data center then broadcasts these weights to the clients, which proceed to train all *consecutive* layers on their datasets, keeping the matched federated layers *frozen*. This procedure is then repeated up to the last layer for which we conduct a weighted averaging based on the class proportions of data points per client. We summarize our Federated Matched Averaging (FedMA) in Algorithm 1. The FedMA approach requires communication rounds equal to the number of layers in a network. In Figure 1 we show that with layer-wise matching FedMA performs well on the deeper VGG-9 CNN as well as LSTMs. In the more challenging heterogeneous setting, FedMA outperforms FedAvg, FedProx trained with same number of communication rounds (4 for LeNet and LSTM and 9 for VGG-9) and other baselines, i.e. client individual CNNs and their ensemble.

**FedMA with communication**  We've shown that in the heterogeneous data scenario FedMA outperforms other federated learning approaches, however it still lags in performance behind the entire data training. Of course the entire data training is not possible under the federated learning constraints, but it serves as performance upper bound we should strive to achieve. To further improve the performance of our method, we propose *FedMA with communication*, where local clients receive the matched global model at the beginning of a new round and reconstruct their local models with the size equal to the original local models (*e.g.* size of a VGG-9) based on the matching results of the previous round. This procedure allows to keep the size of the global model small in contrast to a naive strategy of utilizing full matched global model as a starting point across clients on every round.

---

**Algorithm 1:** Federated Matched Averaging (FedMA)

---

**Input** : local weights of $N$-layer architectures $\{W_{j,1}, \ldots, W_{j,N}\}_{j=1}^{J}$ from $J$ clients
**Output:** global weights $\{W_1, \ldots, W_N\}$
$n = 1$;
**while** $n \leq N$ **do**
   **if** $n < N$ **then**
      $\{\Pi_j\}_{j=1}^{J} = \text{BBP-MAP}(\{W_{j,n}\}_{j=1}^{J})$ ;      // call BBP-MAP to solve Eq. 2
      $W_n = \frac{1}{J}\sum_j W_{j,n}\Pi_j^T$ ;
   **else**
      $W_n = \sum_{k=1}^{K}\sum_j p_{jk}W_{jl,n}$ where $p_k$ is fraction of data points with label $k$ on worker $j$;
   **end**
   **for** $j \in \{1, \ldots, J\}$ **do**
      $W_{j,n+1} \leftarrow \Pi_j W_{j,n+1}$ ;      // permutate the next-layer weights
      Train $\{W_{j,n+1}, \ldots, W_{j,L}\}$ with $W_n$ frozen;
   **end**
   $n = n + 1$;
**end**

---

## 3 EXPERIMENTS

We present an empirical study of FedMA with communication and compare it with state-of-the-art methods *i.e.* FedAvg (McMahan et al., 2017) and FedProx (Sahu et al., 2018); analyze the performance under the growing number of clients and visualize the matching behavior of FedMA to study its interpretability. Our experimental studies are conducted over three real world datasets. Summary information about the datasets and associated models can be found in supplement Table 3.

**Experimental Setup** We implemented FedMA and the considered baseline methods in PyTorch (Paszke et al., 2017). We deploy our empirical study under a simulated federated learning environment where we treat one centralized node in the distributed cluster as the data center and the other nodes as local clients. All nodes in our experiments are deployed on *p3.2xlarge* instances on Amazon EC2. We assume the data center samples all the clients to join the training process for every communication round for simplicity.

For the CIFAR-10 dataset, we use data augmentation (random crops, and flips) and normalize each individual image (details provided in the Supplement). We note that we ignore all batch normalization (Ioffe & Szegedy, 2015) layers in the VGG architecture and leave it for future work.

For CIFAR-10, we considered two data partition strategies to simulate federated learning scenario: (i) homogeneous partition where each local client has approximately equal proportion of each of the classes; (ii) heterogeneous partition for which number of data points and class proportions are unbalanced. We simulated a heterogeneous partition into $J$ clients by sampling $\mathbf{p}_k \sim \text{Dir}_J(0.5)$ and allocating a $\mathbf{p}_{k,j}$ proportion of the training instances of class $k$ to local client $j$. We use the original test set in CIFAR-10 as our global test set for comparing performance of all methods. For the Shakespeare dataset, we treat each speaking role as a client (Caldas et al., 2018) resulting in a natural heterogeneous partition. We preprocess the Shakespeare dataset by filtering out the clients with less than 10k datapoints and sampling a random subset of $J = 66$ clients. We allocate 80% of the data for training and amalgamate the remaining data into a global test set.

**Communication Efficiency and Convergence Rate** In this experiment we study performance of FedMA with communication. Our goal is to compare our method to FedAvg and FedProx in terms of the total message size exchanged between data center and clients (in Gigabytes) and the number of communication rounds (recall that completing one FedMA pass requires number of rounds equal to the number of layers in the local models) needed for the global model to achieve good performance on the test data. We also compare to the performance of an ensemble method. We evaluate all methods under the heterogeneous federated learning scenario on CIFAR-10 with $J = 16$ clients with VGG-9 local models and on Shakespeare dataset with $J = 66$ clients with 1-layer LSTM net-

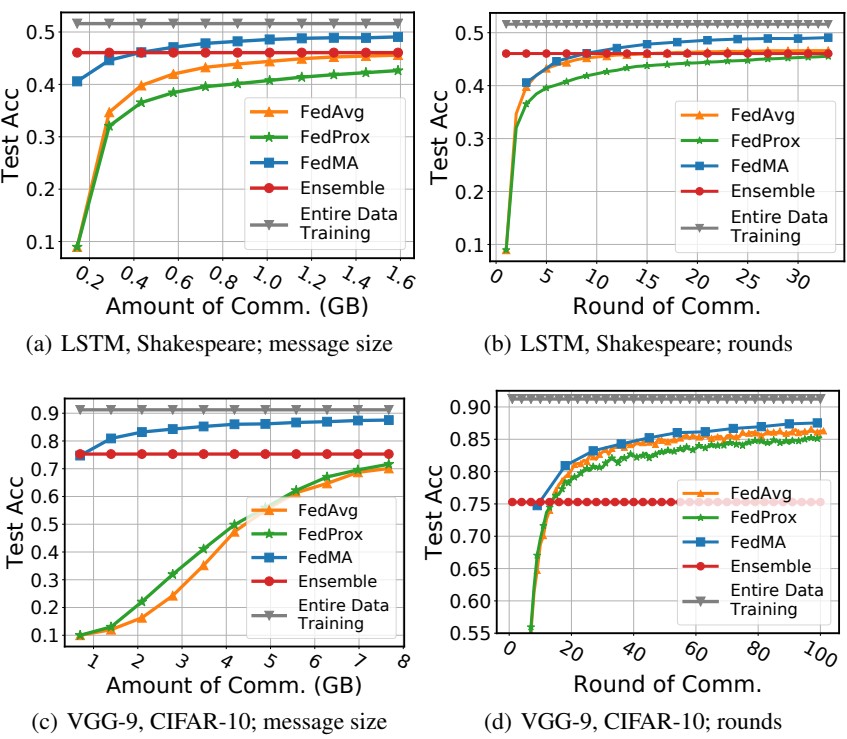

(a) LSTM, Shakespeare; message size      (b) LSTM, Shakespeare; rounds

(c) VGG-9, CIFAR-10; message size      (d) VGG-9, CIFAR-10; rounds

Figure 2: Convergence rates of various methods in two federated learning scenarios: training VGG-9 on CIFAR-10 with $J = 16$ clients and training LSTM on Shakespeare dataset with $J = 66$ clients.

work. We fix the total rounds of communication allowed for FedMA, FedAvg, and FedProx *i.e.* 11 rounds for FedMA and 99/33 rounds for FedAvg and FedProx for the VGG-9/LSTM experiments respectively. We notice that number of local training epochs is a common parameter shared by the three considered methods, we thus tune this parameter (denoted $E$; comprehensive analysis will be presented in the next experiment) and report the convergence rate under $E$ that yields the best final model accuracy over the global test set. We also notice that there is another hyper-parameter in FedProx *i.e.* the coefficient $\mu$ associated with the proxy term, we also tune the parameter using grid search and report the best $\mu$ we found (0.001) for both VGG-9 and LSTM experiments. FedMA outperforms FedAvg and FedProx in all scenarios (Figure 2) with its advantage especially pronounced when we evaluate convergence as a function of the message size in Figures 2(a) and 2(c). Final performance of all trained models is summarized in Tables 1 and 2.

**Effect of local training epochs** As studied in previous work (McMahan et al., 2017; Caldas et al., 2018; Sahu et al., 2018), the number of local training epochs $E$ can affect the performance of FedAvg and sometimes lead to divergence. We conduct an experimental study on the effect of $E$ over FedAvg, FedProx, and FedMA on VGG-9 trained on CIFAR-10 under heterogeneous setup. The candidate local epochs we consider are $E \in \{10, 20, 50, 70, 100, 150\}$. For each of the candidate $E$, we run FedMA for 6 rounds while FedAvg and FedProx for 54 rounds and report the final accuracy that each methods achieves. The result is shown in Figure 3. We observe that training longer benefits FedMA, supporting our assumption that FedMA performs best on local models with higher quality. For FedAvg, longer local training leads to deterioration of the final accuracy, which matches the observations made in the previous literature (McMahan et al., 2017; Caldas et al., 2018; Sahu et al., 2018). FedProx only partially alleviates this problem. The result

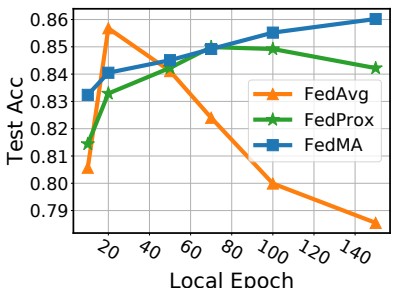

Figure 3: The effect of number of local training epochs on various methods.

of this experiment suggests that FedMA is the only method that local clients can use to train their model as long as they want.

Table 1: Trained models summary for VGG-9 trained on CIFAR-10 as shown in Figure 2

| **Method** | FedAvg | FedProx | Ensemble | FedMA |
|---|---|---|---|---|
| Final Accuracy(%) | 86.29 | 85.32 | 75.29 | **87.53** |
| Best local epoch($E$) | 20 | 20 | N/A | 150 |
| Model growth rate | 1× | 1× | 16× | 1.11× |

Table 2: Trained models summary for LSTM trained on Shakespeare as shown in Figure 2

| **Method** | FedAvg | FedProx | Ensemble | FedMA |
|---|---|---|---|---|
| Final Accuracy(%) | 46.63 | 45.83 | 46.06 | **49.07** |
| Best local epoch($E$) | 2 | 5 | N/A | 5 |
| Model growth rate | 1× | 1× | 66× | 1.06× |

**Handling data bias**  Real world data often exhibit multimodality within each class, *e.g.* geodiversity. It has been shown that an observable amerocentric and eurocentric bias is present in the widely used ImageNet dataset (Shankar et al., 2017; Russakovsky et al., 2015). Classifiers trained on such data "learn" these biases and perform poorly on the under-represented domains (modalities) since correlation between the corresponding dominating domain and class can prevent the classifier from learning meaningful relations between features and classes. For example, classifier trained on amerocentric and eurocentric data may learn to associate white color dress with a "bride" class, therefore underperforming on the wedding images taken in countries where wedding traditions are different (Doshi, 2018).

The data bias scenario is an important aspect of federated learning, however it received little to no attention in the prior federated learning works. In this study we argue that FedMA can handle this type of problem. If we view each domain, e.g. geographic region, as one client, local models will not be affected by the aggregate data biases and learn meaningful relations between features and classes. FedMA can then be used to learn a good global model without biases. We have already demonstrated strong performance of FedMA on federated learning problems with heterogeneous data across clients and this scenario is very similar. To verify this conjecture we conduct the following experiment.

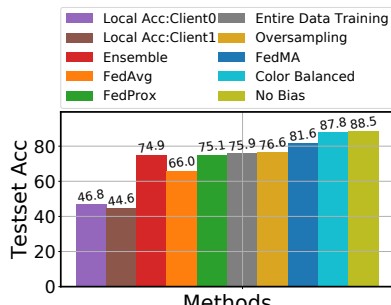

Figure 4:   Performance on skewed CIFAR-10 dataset.

We simulate the skewed domain problem with CIFAR-10 dataset by randomly selecting 5 classes and making 95% training images in those classes to be grayscale. For the remaining 5 we turn only 5% of the corresponding images into grayscale. By doing so, we create 5 grayscale images dominated classes and 5 colored images dominated classes. In the test set, there is half grayscale and half colored images for each class. We anticipate entire data training to pick up the uninformative correlations between greyscale and certain classes, leading to poor test performance without these correlations. In Figure 4 we see that entire data training performs poorly in comparison to the regular (i.e. No Bias) training and testing on CIFAR-10 dataset without any grayscaling. This experiment was motivated by Olga Russakovsky's talk at ICML 2019.

Next we compare the federated learning based approaches. We split the images from color dominated classes and grayscale dominated classes into 2 clients. We then conduct FedMA with communication, FedAvg, and FedProx with these 2 clients. FedMA noticeably outperforms the entire

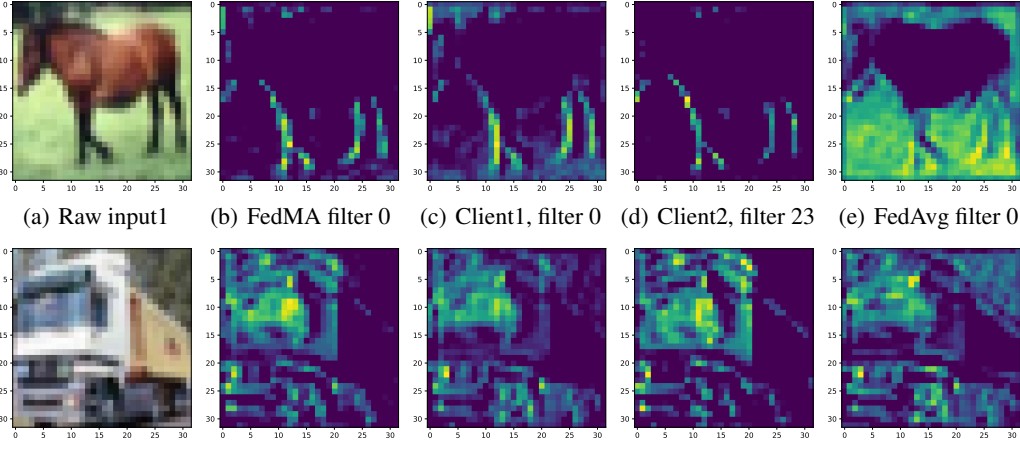

| (a) Raw input1 | (b) FedMA filter 0 | (c) Client1, filter 0 | (d) Client2, filter 23 | (e) FedAvg filter 0 |
| --- | --- | --- | --- | --- |

| (f) Raw input2 | (g) FedMA filter 0 | (h) Client 1, filter0 | (i) Client 2, filter23 | (j) FedAvg filter0 |
| --- | --- | --- | --- | --- |

Figure 5: Representations generated by the first convolution layers of locally trained models, FedMA global model and the FedAvg global model.

data training and other federated learning approach as shown in Figure 4. This result suggests that FedMA may be of interest beyond learning under the federated learning constraints, where entire data training is the performance *upper bound*, but also to eliminate data biases and *outperform* entire data training.

We consider two additional approaches to eliminate data bias without the federated learning constraints. One way to alleviate data bias is to selectively collect more data to debias the dataset. In the context of our experiment, this means getting more colored images for grayscale dominated classes and more grayscale images for color dominated classes. We simulate this scenario by simply doing a full data training where each class in both train and test images has equal amount of grayscale and color images. This procedure, Color Balanced, performs well, but selective collection of new data in practice may be expensive or even not possible. Instead of collecting new data, one may consider oversampling from the available data to debias. In Oversampling, we sample the underrepresented domain (via sampling with replacement) to make the proportion of color and grayscale images to be equal for each class (oversampled images are also passed through the data augmentation pipeline, e.g. random flipping and cropping, to further enforce the data diversity). Such procedure may be prone to overfitting the oversampled images and we see that this approach only provides marginal improvement of the model accuracy compared to centralized training over the skewed dataset and performs noticeably worse than FedMA.

**Data efficiency** It is known that deep learning models perform better when more training data is available. However, under the federated learning constraints, data efficiency has not been studied to the best of our knowledge. The challenge here is that when new clients join the federated system, they each bring their own version of the data distribution, which, if not handled properly, may deteriorate the performance despite the growing data size across the clients. To simulate this scenario we first partition the entire training CIFAR-10 dataset into 5 homogeneous pieces. We then partition each homogeneous data piece further into 5 sub-pieces heterogeneously. Using this strategy, we partition the CIFAR-10 training set into 25 heterogeneous small sub-datasets con-

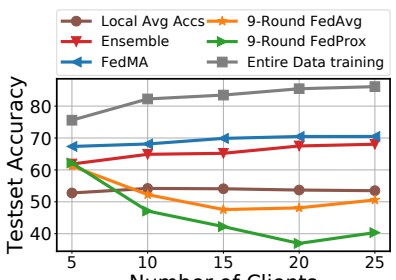

Figure 6: Data efficiency under the increasing number of clients.

taining approximately 2k points each. We conduct a 5-step experimental study: starting from a randomly selected homogeneous piece consisting of 5 associated heterogeneous sub-pieces, we simulate a 5-client federated learning heterogeneous problem. For each consecutive step, we add one of the remaining homogeneous data pieces consisting of 5 new clients with heterogeneous sub-datasets. Results are presented in Figure 6. Performance of FedMA (with a single pass) improves

when new clients are added to the federated learning system, while FedAvg with 9 communication rounds deteriorates.

**Interpretability** One of the strengths of FedMA is that it utilizes communication rounds more efficiently than FedAvg. Instead of directly averaging weights element-wise, FedMA identifies matching groups of convolutional filters and then averages them into the global convolutional filters. It's natural to ask *"How does the matched filters look like?"*. In Figure 5 we visualize the representations generated by a pair of matched local filters, aggregated global filter, and the filter returned by the FedAvg method over the same input image. Matched filters and the global filter found with FedMA are extracting the same feature of the input image, i.e. filter 0 of client 1 and filter 23 of client 2 are extracting the position of the legs of the horse, and the corresponding matched global filter 0 does the same. For the FedAvg, global filter 0 is the average of filter 0 of client 1 and filter 0 of client 2, which clearly tampers the leg extraction functionality of filter 0 of client 1.

## 4  CONCLUSION

We presented Federated Matched Averaging (FedMA), a layer-wise federated learning algorithm designed for modern CNNs and LSTMs architectures that accounts for permutation invariance of the neurons and permits global model size adaptation. Our method significantly outperforms prior federated learning algorithms in terms of its convergence when measured by the size of messages exchanged between server and the clients during training. We demonstrated that FedMA can efficiently utilize well-trained local modals, a property desired in many federated learning applications, but lacking in the prior approaches. We have also presented an example where FedMA can help to resolve some of the data biases and outperform aggregate data training.

In the future work we want to extend FedMA to improve federated learning of LSTMs using approximate quadratic assignment solutions from the optimal transport literature, and enable additional deep learning building blocks, *e.g.* residual connections and batch normalization layers. We also believe it is important to explore fault tolerance of FedMA and study its performance on the larger datasets, particularly ones with biases preventing efficient training even when the data can be aggregated, *e.g.* Inclusive Images (Doshi, 2018).

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

## A  SUMMARY OF THE DATASETS USED IN THE EXPERIMENTS

The details of the datasets and hyper-parameters used in our experiments are summarized in Table 3. In conducting the "freezing and retraining" process of FedMA, we notice when retraining the last FC layer while keeping all previous layers frozen, the initial learning rate we use for SGD doesn't lead to a good convergence (this is only for the VGG-9 architecture). To fix this issue, we divide the initial learning rate by 10 *i.e.* using $10^{-4}$ for the last FC layer retraining and allow the clients to retrain for 3 times more epochs. We also switch off the $\ell_2$ weight decay during the "freezing and retraining" process of FedMA except for the last FC layer where we use a $\ell_2$ weight decay of $10^{-4}$. For language task, we observe SGD with a constant learning rate works well for all considered methods.

In our experiments, we use FedAvg and FedProx variants without the shared initialization since those would likely be more realistic when trying to aggregate locally pre-trained models. And FedMA still performs well in practical scenarios where local clients won't be able to share the random initialization.

Table 3: The datasets used and their associated learning models and hyper-parameters.

| **Method** | MNIST | CIFAR-10 | Shakespeare (McMahan et al., 2017) |
|---|---|---|---|
| # Data points | $60,000$ | $50,000$ | $1,017,981$ |
| Model | LeNet | VGG-9 | LSTM |
| # Classes | 10 | 10 | 80 |
| # Parameters | 431k | $3,491$k | 293k |
| Optimizer | SGD | | SGD |
| Hyper-params. | Init lr: 0.01, 0.001 (last layer) | | lr: 0.8(const) |
| | momentum: 0.9, $\ell_2$ weight decay: $10^{-4}$ | | |

## B  DETAILS OF MODEL ARCHITECTURES AND HYPER-PARAMETERS

The details of the model architectures we used in the experiments are summarized in this section. Specifically, details of the VGG-9 model architecture we used can be found in Table 4 and details of the 1-layer LSTM model used in our experimental study can be found in Table 5.

## C  DATA AUGMENTATION AND NORMALIZATION DETAILS

In preprocessing the images in CIFAR-10 dataset, we follow the standard data augmentation and normalization process. For data augmentation, random cropping and horizontal random flipping are used. Each color channels are normalized with mean and standard deviation by $\mu_r = 0.491372549, \mu_g = 0.482352941, \mu_b = 0.446666667, \sigma_r = 0.247058824, \sigma_g = 0.243529412, \sigma_b = 0.261568627$. Each channel pixel is normalized by subtracting the mean value in this color channel and then divided by the standard deviation of this color channel.

## D  EXTRA EXPERIMENTAL DETAILS

### D.1  SHAPES OF FINAL GLOBAL MODEL

Here we report the shapes of final global VGG and LSTM models returned by FedMA with communication.

Table 4: Detailed information of the VGG-9 architecture used in our experiments, all non-linear activation function in this architecture is ReLU; the shapes for convolution layers follows $(C_{in}, C_{out}, c, c)$

| Parameter | Shape | Layer hyper-parameter |
|---|---|---|
| layer1.conv1.weight | $3 \times 32 \times 3 \times 3$ | stride:1;padding:1 |
| layer1.conv1.bias | 32 | N/A |
| layer2.conv2.weight | $32 \times 64 \times 3 \times 3$ | stride:1;padding:1 |
| layer2.conv2.bias | 64 | N/A |
| pooling.max | N/A | kernel size:2;stride:2 |
| layer3.conv3.weight | $64 \times 128 \times 3 \times 3$ | stride:1;padding:1 |
| layer3.conv3.bias | 128 | N/A |
| layer4.conv4.weight | $128 \times 128 \times 3 \times 3$ | stride:1;padding:1 |
| layer4.conv4.bias | 128 | N/A |
| pooling.max | N/A | kernel size:2;stride:2 |
| dropout | N/A | $p = 5\%$ |
| layer5.conv5.weight | $128 \times 256 \times 3 \times 3$ | stride:1;padding:1 |
| layer5.conv5.bias | 256 | N/A |
| layer6.conv6.weight | $256 \times 256 \times 3 \times 3$ | stride:1;padding:1 |
| layer6.conv6.bias | 256 | N/A |
| pooling.max | N/A | kernel size:2;stride:2 |
| dropout | N/A | $p = 10\%$ |
| layer7.fc7.weight | $4096 \times 512$ | N/A |
| layer7.fc7.bias | 512 | N/A |
| layer8.fc8.weight | $512 \times 512$ | N/A |
| layer8.fc8.bias | 512 | N/A |
| dropout | N/A | $p = 10\%$ |
| layer9.fc9.weight | $512 \times 10$ | N/A |
| layer9.fc9.bias | 10 | N/A |

Table 5: Detailed information of the LSTM architecture in our experiment

| Parameter | Shape |
|---|---|
| encoder.weight | $80 \times 8$ |
| lstm.weight.ih.l0 | $1024 \times 8$ |
| lstm.weight.hh.l0 | $1024 \times 256$ |
| lstm.bias.ih.l0 | 1024 |
| lstm.bias.hh.l0 | 1024 |
| decoder.weight | $80 \times 256$ |
| decoder.bias | 80 |

Table 6: Detailed information of the LSTM architecture in our experiment

| Parameter | Shape | Growth rate (#global / #original params) |
|---|---|---|
| encoder.weight | $80 \times 21$ | $2.63 \times (1,680/640)$ |
| lstm.weight.ih.l0 | $1028 \times 21$ | $2.64 \times (21,588/8,192)$ |
| lstm.weight.hh.l0 | $1028 \times 257$ | $1.01 \times (264,196/262,144)$ |
| lstm.bias.ih.l0 | $1028$ | $1.004 \times (1,028/1,024)$ |
| lstm.bias.hh.l0 | $1028$ | $1.004 \times (1,028/1,024)$ |
| decoder.weight | $80 \times 257$ | $1.004 \times (20,560/20,480)$ |
| decoder.bias | $80$ | $1\times$ |
| Total Number of Parameters | $310,160$ | $1.06\times (310,160/293,584)$ |

Table 7: Detailed information of the final global VGG-9 model returned by FRB; the shapes for convolution layers follows $(C_{in}, C_{out}, c, c)$

| Parameter | Shape | Growth rate (#global / #original params) |
|---|---|---|
| layer1.conv1.weight | $3 \times 47 \times 3 \times 3$ | $1.47 \times (1,269/864)$ |
| layer1.conv1.bias | $47$ | $1.47 \times (47/32)$ |
| layer2.conv2.weight | $47 \times 79 \times 3 \times 3$ | $1.81 \times (33,417/18,432)$ |
| layer2.conv2.bias | $79$ | $1.23 \times (79/64)$ |
| layer3.conv3.weight | $79 \times 143 \times 3 \times 3$ | $1.38 \times (101,673/73,728)$ |
| layer3.conv3.bias | $143$ | $1.12 \times (143/128)$ |
| layer4.conv4.weight | $143 \times 143 \times 3 \times 3$ | $1.24 \times (184,041/147,456)$ |
| layer4.conv4.bias | $143$ | $1.12 \times (143/128)$ |
| layer5.conv5.weight | $143 \times 271 \times 3 \times 3$ | $1.18 \times (348,777/294,912)$ |
| layer5.conv5.bias | $271$ | $1.06 \times (271/256)$ |
| layer6.conv6.weight | $271 \times 271 \times 3 \times 3$ | $1.12 \times (660,969/589,824)$ |
| layer6.conv6.bias | $271$ | $1.06 \times (271/256)$ |
| layer7.fc7.weight | $4336 \times 527$ | $1.09 \times (2,285,072/2,097,152)$ |
| layer7.fc7.bias | $527$ | $1.02 \times (527/512)$ |
| layer8.fc8.weight | $527 \times 527$ | $1.05\times, (277,729/262,144)$ |
| layer8.fc8.bias | $527$ | $1.02 \times (527/512)$ |
| layer9.fc9.weight | $527 \times 10$ | $1.02 \times (5,270/5,120)$ |
| layer9.fc9.bias | $10$ | $1\times$ |
| Total Number of Parameters | $3,900,235$ | $1.11\times (3,900,235/3,491,530)$ |

## D.2 HYPER-PARAMETERS FOR BBP-MAP

We follow FPNM (Yurochkin et al., 2019b) to choose the hyper-parameters of BBP-MAP, which controls the choices of $\theta_i$, $\epsilon$, and the $f(\cdot)$ as discussed in Section 2. More specifically, there are three parameters to choose *i.e.* 1) $\sigma_0^2$, the prior variance of weights of the global neural network; 2) $\gamma_0$, which controls discovery of new hidden states. Increasing $\gamma_0$ leads to a larger final global model; 3) $\sigma^2$ is the variance of the local neural network weights around corresponding global network weights. We empirically analyze the different choices of the three hyper-parameters and find the choice of

$\gamma_0 = 7, \sigma_0^2 = 1, \sigma^2 = 1$ for VGG-9 on CIFAR-10 dataset and $\gamma_0 = 10^{-3}, \sigma_0^2 = 1, \sigma^2 = 1$ for LSTM on Shakespeare dataset lead to good performance in our experimental studies.

## E  PRACTICAL CONSIDERATIONS

Following from the discussion in PFNM, here we briefly discuss the time complexity of FedMA. For simplicity, we focus on a single-layer matching and assume all participating clients train the same model architecture. The complexity for matching the entire model follows trivially from this discussion. The worst case complexity is achieved when no hidden states are matched and is equal to $\mathcal{O}(D \cdot (JL)^2)$ for building the cost matrix and $\mathcal{O}((JL)^3)$ for running the Hungarian algorithm where the definitions of $D, J$, and $L$ follow the discussion in Section 2. The best complexity per layer is (achieved when all hidden states are matched) $\mathcal{O}(D \cdot L^2 + L^3)$. Practically, when the number of participating clients *i.e.* $J$ is large and each client trains a big model, the speed of our algorithm can be relatively slow.

To seed up the Hungarian algorithm. Although there isn't any algorithm that achieves lower complexity, better implementation improves the constant significantly. In our experiments, we used an implementation based on shortest path augmentation *i.e.* `lapsolver` [2]. Empirically, we observed that this implementation of the Hungarian algorithm leads to orders of magnitude speed ups over the vanilla implementation.

## F  HYPER-PARAMETERS FOR THE HANDLING DATA BIAS EXPERIMENTS

In conducting the "handling data bias" experiments. We re-tune the local epoch $E$ for both FedAvg and FedProx. The considered candidates of $E$ are $\{5, 10, 20, 30\}$. We observe that a relatively large choice of $E$ can easily lead to poor convergence of FedAvg. While FedProx tolerates larger choices of $E$ better, a smaller $E$ can always lead to good convergence. We use $E = 5$ for both FedAvg and FedProx in our experiments. For FedMA, we choose $E = 50$ since it leads to a good convergence. For the "oversampling" baseline, we found that using SGD to train VGG-9 over oversampled dataset doesn't lead to a good convergence. Moreover, when using constant learning rate, SGD can lead to model divergence. Thus we use AMSGRAD (Reddi et al., 2018) method for the "oversampling" baseline and train for 800 epochs. To make the comparison fair, we use AMSGRAD for all other centralized baselines to get the reported results in our experiments. Most of them converges when training for 200 epochs. We also test the performance of the "Entire Data Training", "Color Balanced", and "No Bias" baselines over SGD. We use learning rate $0.01$ and $\ell_2$ weight decay at $10^{-4}$ and train for 200 epochs for those three baselines. It seems the "Entire Data Training" and "No Bias" baselines converges to a slightly better accuracy *i.e.* $78.71\%$ and $91.23\%$ respectively (compared to $75.91\%$ and $88.5\%$ for AMSGRAD). But the "Color Balanced" doesn't seem to converge better accuracy (we get $87.79\%$ accuracy for SGD and $87.81\%$ for AMSGRAD).

---

[2]`https://github.com/cheind/py-lapsolver`

