# OpenReview forum: "Federated Learning with Matched Averaging"
_ICLR.cc/2020/Conference — Accept (Talk)_

### Official Review · AnonReviewer1 · 2019-10-22
**Official Blind Review #1**

**Rating:** 8

**Review:**

Post Rebuttal Summary
---------------------------------
I have nudged my score up to an "Accept", based on my comments to the rebuttal below. I hope the authors continue to improve the readability of Sec. 2.1

Review Summary
--------------
Overall I think this is almost above the bar to be accepted, and I could be persuaded with a strong rebuttal.  The strengths here are the extensive experiments and the easy-to-implement method. The primary weakness of this paper is that it is a "straightfoward" way to extend the BBP-MAP method to CNNs and RNNs, so the methodological novelty is weak relative to the BBP-MAP past work (Yurochkin et al. ICML 2019). Other technical weaknesses limit the ability to use this method on clients with diverse class distributions, which will be common in real deployments.

Paper Summary
-------------
This paper addresses the problem of federated learning, where J separate "clients" with disjoint datasets each train a neural network model for a supervised problem, and then try to aggregate all J individual client models into one "global model" in a coherent way. The natural problem is that due to hidden units being permutable within one network, naively taking parameter averages across two client models will lead to bad accuracy without first coming up with a consistent ordering of the units in each layer.

Previous work (Yurochkin et al. ICML 2019) has developed a Bayesian nonparametric model based on the Beta-Bernoulli Process (BBP) for the case of federated learning of multi-layer perceptrons. However, the extension to convolutional layers or recurrent layers has yet to be solved, which is the focus of this paper.

This paper's algorithm (Federated Matched Averaging (FedMA), see Alg 1), proceeds by iteratively stepping thru the CNN or RNN layer by layer greedily from input to output. At each layer, we first solve a BBP-MAP optimization (bipartite matching using a BBP maximum a-posteriori objective as cost function, a subprocedure taken direclty from Yurochkin et al.). This obtains a consistent low-cost permutation for each client model. Then, the global model weights for that layer is the average of the aligned client weights. After the current layer update, each client keeps training, keeping all layers up to the current frozen but revising later layers. This layer-by-layer training can be applied to both CNNs and RNNs.

The proposed approach is compared to FedAvg and FedProx on MNIST and CIFAR image classification tasks with CNNs, and Shakespeare text classification tasks with RNNs. Later experiments explore the effect of communication efficiency (MB transfered between client and master), effect of local training epochs, handling biased class distributions, and interpretabilty.




Novelty & Significance
-----------------------
Solving federated learning problems is of increasing practical importance, and certainly trying to do so for CNNs and RNNs (more than just large MLPs) is important. So I like where the paper is going.

Although the method is "new", it is more or less a straightforward extension of work by Yurochkin et al. (ICML 2019) to CNNs and RNNs. If you read the last few sentences of Yurochkin et al., you'll see "Finally, it is of interest to extend our model-ing framework to other architectures such as Convolutional Neural Networks (CNNs) and Recurrent Neural Networks (RNNs). The permutation invariance necessitating matching inference also arises in CNNs since any permutation of the filters results in the same output, however additional bookkeeping is needed due to the pooling operations." I view this paper as a well-executed implementation of this "bookkeeping". Certainly not trivial, but to some readers perhaps not clearly "above the bar" for a top conference like ICLR.


Technical Concerns
------------------

## Concern 1: Client models will not always be alignable after permutation

My first concern is that there will not always be a one-to-one permutation of the neurons learned by two client models with different class distributions. Given fixed capacity at each layer, some clients may learn a filter for "horse hooves" (esp. if horse images are common to that client), while other clients may learn a filter for "snake skin" (if snakes are more common to that client). I wonder if we can quantify how well the aligned filters match in practice, and if there is any benefit to revising the alignment to allow some client-specific customizations (e.g. by having the global model can learn more units than the client model).

## Concern 2: Use of the BBP-MAP subprocedure poorly motivated

The paper prioritizes a clean and easy-to-implement algorithm to resolve practical alignment issues between client CNN and RNN models. However, I was a bit underwhelmed that the BBP-MAP solution used by Yurochkin et al. was treated as a black-box subprocedure without much justification. I could see 2 preferable alternatives to the current use of BBP-MAP. Either a simpler approach using Eq. 2 with a squared error cost and the Munkres algorithm to solve bipartitite matching to obtain the permutation (which seems more in spirit of the rest of the paper). Or, a more sophisticated probabilistic approach (taking a Bayesian hierarchical model from Yurochkin et al. seriously and forming the estimated global weights from a weighted sums that includes both the clients (weighted by dataset size) and the assumed prior). As it is, I feel the BBP-MAP subprocedure in the current Algorithm 1 is poorly motivated for the task at hand.



Experimental Evaluation
-----------------------

Overall the experiments were extensive and demonstrated several apparent advantages (reduced need to transfer large memory during communication, etc.).


Minor Presentation Concerns
---------------------
Before Eq. 2, you should introduce the "\theta" notation

I'm a bit confused about how "FedMA" differs from "FedMA with communication", even after reading Sec. 2.3. How exactly are communicate costs kept down? What are you sending from master to client at beginning of every "round" if not the full global model (all weights of the CNN)?

**Experience Assessment:**

I have read many papers in this area.

**Review Assessment: Checking Correctness Of Derivations And Theory:**

I assessed the sensibility of the derivations and theory.

**Review Assessment: Checking Correctness Of Experiments:**

I assessed the sensibility of the experiments.

**Review Assessment: Thoroughness In Paper Reading:**

I read the paper at least twice and used my best judgement in assessing the paper.

---

> ### Author Response · Authors · 2019-11-12
> **Response to Reviewer 1**
>
> We thank the reviewer for the thorough review and feedback. We address the concerns raised below.
>
> We extended paragraph "Solving matched averaging" in Section 2.1 clarifying how our approach can learn the size of the global model and the motivation for using BBP-MAP. We hope this addresses both of the technical concerns. To summarize, our approach does allow global model to learn more units than the client models and in your example with "horse hooves" and "snake skin" it will do the right thing, i.e. it will not match "horse hooves" to "snake skin" and instead increase the size of the global model keeping both. In our experiments, when simulating heterogeneous CIFAR-10 partitioning, we use Dirichlet distribution with a small concentration parameter to partition each of the class examples across clients - this results in very diverse class distributions across clients (some might even completely lack several classes). Please see the "Experimental Setup" paragraph in Section 3 for details. Our experiments demonstrate that FedMA performs well under the diverse class distributions scenario. The global neural net found by FedMA is bigger than local models, but only mildly - please see rows "Model growth rate" in Tables 1 and 2.
>
> In our extended "Solving matched averaging" paragprah we also gave a general recipe for performing matched averaging with adaptive global model size. The idea is to consider an extended cost matrix and iteratively apply the Hungarian (Munkres) algorithm. Iterations are needed to handle multiple unknown permutation matrices, but if we only have two neural networks, then a single run of the Hungarian algorithm with our cost matrix is sufficient. We also clarified the motivation for using BBP-MAP: it simply gives us a way to pick cost matrix, matching threshold and model size penalty simultaneously, based on the model of Yurochkin et al. Otherwise, their algorithm is a special case of our framework.
>
> >>> Regarding the novelty
>
> We believe our work has both practical and methodological contributions in comparison to Yurochkin et al. We would like to emphasize that LSTMs matching presented in this paper is special and differs from MLPs and CNNs. In eq. (6) we show that it leads to a quadratic assignment problem due to permutation applied on both sides of the hidden-to-hidden weights in the LSTM cell. Our solution is to use linear assignment corresponding to input-to-hidden weights to find the permutations, but account for the special permutation structure of the hidden-to-hidden weights when averaging them. For the CNNs, we showed that it is essentially same as the MLPs and we agree that in this case it is a relatively straightforward extension of Yurochkin et al. Overall, we think that combining and formalizing permutation invariance structure of all key architectures in the language of permutation matrices (instead of Bayesian modeling) is also a valuable contribution. For example, our formalism shows an interesting connection to Optimal Transport, i.e. eq. (2) is very similar to Wasserstein barycenter formulation, while the quadratic assignment arising in LSTMs is related to Gromov-Wasserstein barycenters. This connection may lead to better estimation of permutations for matched averaging, replacing BBP-MAP.
>
> Another methodological contribution of our work is the combination of layer-wise matching and local re-training. In Figure 1 we experimentally showed that simply extending approach of Yurochkin et al. to CNNs (labeled One-Shot Matching on the plots) only works for basic architectures. FedMA enables efficient federated learning of modern architectures and demonstrates strong empirical performance on more challenging datasets in comparison to Yurochkin et al.
>
> >>> Minor Presentation Concerns
>
> We have added theta definition before the equation.
>
> One round of FedMA requires communication rounds equal to the number of layers in a network. At each of these communications, clients send weights of a single layer, then master matches and averages them and broadcasts back the resulting global weights for this one layer. To summarize, one round of FedMA requires "number of layers" communications, but the total message size is equal to one communication round of FedAvg, i.e. size of the full model. FedMA with communication is basically repeating rounds of FedMA with one important detail. Recall that FedMA learns the global model size, hence the global neural net is usually slightly bigger than the local models. To keep the size of the local models constant when proceeding to the next FedMA round, we re-set local models to the "subsets" of the global model that they were matched to. This has no significant communication overhead as permutation matrices needed to obtain those subsets can be easily stored and broadcasted as lists of integers when running a FedMA round, and each client naturally has a global model copy by the end of each FedMA round.

---

> > ### Comment · AnonReviewer1 · 2019-11-13
> > **Thanks for rebuttal!**
> >
> > I appreciate the careful reply and revisions of Sec 2.1 especially. I think the novelty is above-the-bar (esp if connections to optimal transport could be made more explicit). My technical concern #1 is resolved thanks to a clear response from authors that the method handles the concern. My concern #2 (about motivation for BBP-MAP) is mostly resolved but for some presentation issues (I think perhaps another few editing passes to make sure the presentation of BBP-MAP in Sec 2.1 is clear and well-motivated might be needed, because it still feels like the current simplicity-focused justification is a bit weak).
> >
> > I'm happy to accept the paper.

---

> > > ### Author Response · Authors · 2019-11-14
> > > **We thank Reviewer 1**
> > >
> > > We really appreciate your careful review and encouraging response to our rebuttal. We will elaborate on the connection to optimal transport that we mentioned in our response in the final version of the paper. We will also more smoothly integrate the section on “Solving matched averaging” (including the presentation of BBP-MAP) into the flow of the paper.

---

### Official Review · AnonReviewer2 · 2019-10-23
**Official Blind Review #2**

**Rating:** 8

**Review:**

Edit: Thanks for the thorough and responsive rebuttal! I'm particularly happy to see the additional background on BBP-MAP and the baselines you've added for handling data bias. You've comprehensively addressed my questions and I think this paper should be accepted.

Original review:

The authors extend the recently proposed Probabilistic Federated Neural Matching (PFNM) algorithm of Yurochkin et al. ( 2019) to more kinds of neural networks, show that it isn't as effective for larger models as it is for LeNet-sized ones, and propose enhancements that lead to a state-of-the-art approach they call FedMA. I'm convinced that this represents a meaningful advance in federated learning, although the paper could use some tightening up, and the experiments are somewhat limited.

Some feedback:

- I'd like to see a little bit more description of BBP-MAP, as even though it's not one of the components of the algorithm you directly modify it's still the underlying mathematical primitive. How far is it from having the same effect that the "best possible" permutation would? How is it able to allow the number of neurons in the federated model to grow relative to the size of the client models?

- Can you include the "entire data" baseline in more of the figures/plots (especially Figure 2)?

- The models and datasets covered in the experiments are adequate to demonstrate that the presented technique is worth exploring, but probably not for someone considering applying it in the context of a deployed federated learning application. Since federated learning is a problem domain motivated more by applied concerns (privacy, edge vs. cloud compute, on-device ML) than other areas of machine learning theory, it would be particularly valuable to see experiments at larger scale (in particular, on larger or more realistic datasets).

- The section that demonstrates how your model addresses skewed data domains is fascinating! That's one area in which your experiments are directly relevant to federated learning in practice, and it's a rapidly growing area of research in itself (e.g. in its relationship to causal learning that Leon Bottou has recently been exploring). Exploring this further could make for a whole separate paper. In the mean time, though, is there some kind of equivalent of the "entire data" baseline that would represent e.g. the best known technique for taking into account skewed domains outside the federated context?

**Experience Assessment:**

I do not know much about this area.

**Review Assessment: Checking Correctness Of Derivations And Theory:**

I assessed the sensibility of the derivations and theory.

**Review Assessment: Checking Correctness Of Experiments:**

I assessed the sensibility of the experiments.

**Review Assessment: Thoroughness In Paper Reading:**

I read the paper at least twice and used my best judgement in assessing the paper.

---

> ### Author Response · Authors · 2019-11-12
> **Response to Reviewer 2**
>
> We thank the reviewer for the feedback and provide answers to the raised concerns below.
>
> >>> Additional details on BBP-MAP and adaptive global model size
>
> We extended Section 2.1 with a general procedure for matched averaging with adaptive global model size, where BBP-MAP can be seen as a specific way to carry out the optimization. The idea behind the adaptive global model size is to introduce additional columns in the cost matrix to avoid "poor" matches while penalizing the model size. Please see eq. (3) and the surrounding discussion.
> Regarding the "best possible" permutation, for two neural nets Hungarian algorithm will find the global optima, but when averaging multiple neural nets, the iterative procedure we describe is only guaranteed to find a local optima.
>
> >>> Can you include the "entire data" baseline in more of the figures/plots (especially Figure 2)?
>
> We included the “entire data” baseline in all figures (except Figure 3, where it is not applicable).
>
> >>> The models and datasets covered in the experiments are adequate to demonstrate that the presented technique is worth exploring, but probably not for someone considering applying it in the context of a deployed federated learning application.
>
> We agree that the scale of our current experiments is lagging behind the size of the real world federated learning applications. However, as federated learning is a relatively new problem in the literature, we believe it is also an issue in the majority of the prior results in this area (e.g. several papers studied federated learning simulated with CIFAR-10 as we did, but we are not aware of any paper with ImageNet experiments). We note that there in an optimism that FedMA will benefit from larger datasets: our "Data efficieny" experiment (Figure 5) shows that FedMA utilizes additional data more efficiently in comparison to other federated learning approaches. Further, the "Effect  of local training epochs" (Figure 3) experiment shows that FedMA is the only method truly benefiting from well-trained local models, which might be important as we move onto larger datasets.
>
> For the future work, we are exploring potential large scale datasets representative of the practical federated learning and planning to consider federated learning experiments simulated from ImageNet.
>
> >>> Is there some kind of equivalent of the "entire data" baseline that would represent e.g. the best known technique for taking into account skewed domains outside the federated context?
>
> We agree that this is a valuable and natural question. We added three additional baselines to Figure 4 in the updated manuscript and updated the corresponding "Handling data bias" paragraph in the draft.
>
> i) Vanilla VGG training over CIFAR-10. For this baseline (No Bias), we simply conduct normal model training over the entire CIFAR-10 dataset without any grayscaling. This baseline is not a realistic solution to the data bias and is simply added for the reference.
>
> ii) One way to alleviate data bias is to selectively collect more data to debias the dataset. In the context of our experiment, this means getting more colored images for grayscale dominated classes and more grayscale images for color dominated classes. To simulate this scenario we simply do a full data training where each class in both train and test images has equal amount of grayscale and color images. This procedure, Color Balanced, performs well, but selective collection of new data in practice may be expensive or even not possible.
>
> iii) Instead of collecting new data, one may consider oversampling from the available data to debias. In Oversampling, we sample the underrepresented domain (via sampling with replacement) to make the proportion of color and grayscale images to be equal for each class (oversampled images are also passed through the data augmentation pipeline, e.g. random flipping and cropping, to further enforce the data diversity). Such procedure may be prone to overfitting the oversampled images and we see that this approach only provides marginal improvement of the model accuracy compared to centralized training over the skewed dataset and performs noticeably worse than FedMA.
>
> To conclude, we agree that learning with skewed data is an exciting research direction. Our preliminary experiment indicates that FedMA has the potential to mitigate the data skewness, but further work is needed to obtain solutions as good as the one corresponding to the balanced data training (and without expensive additional data collection).

---

### Official Review · AnonReviewer3 · 2019-11-06
**Official Blind Review #3**

**Rating:** 6

**Review:**

This paper offers a beautiful and simple method for federated learning. Strong empirical results.

Important area.
                                                                                                                                                                                                                                                                                                                                                                                                .

**Experience Assessment:**

I have read many papers in this area.

**Review Assessment: Checking Correctness Of Derivations And Theory:**

I assessed the sensibility of the derivations and theory.

**Review Assessment: Checking Correctness Of Experiments:**

I carefully checked the experiments.

**Review Assessment: Thoroughness In Paper Reading:**

I made a quick assessment of this paper.

---

> ### Public Comment · ~Anthony_Wittmer1 · 2019-11-06
> **More details**
>
> Hi, could you please provide more details for the review?

---

> ### Public Comment · ~Nan_Jiang7 · 2019-11-08
> **rude review**
>
> I have never seen such a review. This is rude, How can you determine the quality of this paper based on "important area".

---

### Author Response · Authors · 2019-11-12
**General Response**

We thank all the reviewers for the thoughtful comments. We have followed the reviewers suggestions to revise and improve our manuscript, while providing extra experiments as requested. One notable addition is the extended paragraph "Solving matched averaging" in Section 2.1 describing a general recipe for performing the matched averaging of neural nets with adaptive size of the global neural net. BBP-MAP then can be seen as a specific way of carrying out the optimization procedure we described. We answer each reviewer’s questions individually.

---

### Public Comment · ~Martin_Jaggi1 · 2020-01-07
**relation to 'Model Fusion via Optimal Transport'**

dear authors
congrats on your nicely written paper on this cool application!
we have a similar&simultaneous approach in the NeurIPS 2019 optimal transport workshop https://arxiv.org/abs/1910.05653 , where we also considered federated learning as an application. we also added some additional baselines for the standalone merging/fusion operator.
would you mind adding it to the discussion for your camera ready version?
thanks in advance!
martin & sidak

---

> ### Author Response · Authors · 2020-01-26
> **Relation of OT fusion to prior work**
>
> Hi Martin & Sidak,
>
> Thank you for bringing your work to our attention. We will add OT fusion to the discussion in the camera ready version of our paper. We also wish to mention that there is a prior work [1] that proposed a similar "align and average" framework that we build on in FedMA. We would appreciate if you can add discussion of [1] and FedMA to your paper.
>
> Best regards,
> Authors
>
> [1] M. Yurochkin et al, Bayesian Nonparametric Federated Learning of Neural Networks, ICML 2019.

---

### Decision · Program_Chairs · 2019-12-19

**Decision:**

Accept (Talk)

**Comment:**

The authors presented a Federate Learning algorithm which constructs the global model layer-wise by matching and averaging hidden representations. They empirically demonstrate their method outperforms existing federated learning algorithms

This paper has received largely positive reviews. Unfortunately one reviewer wrote a very short review but was generally appreciative of the work. Fortunately, R1 wrote a detailed review with very specific questions and suggestions. The authors have addresses most of the concerns of the reviewers and I have no hesitation in recommending that this paper should be accepted. I request the authors to incorporate all suggestions made by the reviewers.